# Dynamic livelihood impacts of COVID-19 on different rural households in mountainous areas of China

**Chengchao Wang** [1,2]*, **Xiu He** [2], **Xianqiang Song** [1], **Shanshan Chen** [2], **Dongshen Luo** [2]

**1** School of Environmental and Chemical Engineering, Foshan University, Foshan, P. R. China, **2** Key Laboratory for Humid Subtropical Eco-geographical Processes of the Ministry of Education, Fujian Normal University, Fuzhou, P. R. China

* wchc79@163.com

## Abstract

### Background

The outbreak of COVID-19 pandemic has brought about severe negative livelihood consequences for rural households worldwide. However, the heterogeneity and dynamics of livelihood impacts have been under-researched. There is also lacking a livelihood assessment of the pandemic based on a whole pandemic cycle. This study aimed to investigate the dynamic and heterogeneous livelihood impacts of COVID-19 pandemic for rural households in 2020 based on a case study of Southeast China.

### Methods

The pandemic in China had experienced a complete cycle from initial outbreak, to intermediate recovery and finally new normal stage in 2020. We conducted face-to-face interviews with 95 rural households randomly drawn from 2 rural villages in *Xunwu County*, *Jiangxi Province*, Southeast China. The sampled households are interviewed with a questionnaire through face-to-face surveys in February and March, 2021 to evaluate the overall livelihood impacts of the pandemic during 2020. The survey collected data on demographic and economic characteristics, governmental control measures, and effects of the COVID-19 on agricultural production, employment, income, education, and daily life. In-depth interviews are also conducted to clarify the livelihood impacts of COVID-19 on villages.

### Results

Results showed that the pandemic tremendously caused substantially negative livelihood impacts, including decreasing household income, and disorders in daily lives. The average income loss of all survey households is 6,842 RMB, accounting for 13.01% of the total household income in 2020. Containment measures also resulted in a series of disturbances in daily lives, such as rising food price additional expenditures, travel restrictions, party restrictions, closure of schools and deceasing living standards. There is remarkable household heterogeneity in the livelihood impacts. Results also revealed that the livelihood

**Data Availability Statement:** All relevant data are within the paper and its Supporting Information files.

**Funding:** This research was jointly funded by the National Natural Science Foundation of China (Grant No. 4217011056) and Humanity and Social Science Youth Foundation of Chinese Ministry of Education (Grant No. 20YJCZH112). The funders had no role in study design, data collection and analysis, decision to publish, or preparation of the manuscript. The authors received no specific funding for this work.

**Competing interests:** The authors have declared that no competing interests exist.

strategies of rural households to cope with the threat of COVID-19 were different in various pandemic stages.

## Conclusion

Our findings have illustrated the severity and heterogeneity of livelihood impacts on rural households induced by COVID-19 pandemic. The dynamics of livelihood impacts is also highlighted in the study. Several policy suggestion was proposed to mitigate these negative consequences of the pandemic.

## Introduction

Originating from the wildlife trade and emerging as a global pandemic, COVID-19 has been a great threat to lives and livelihoods of worldwide communities [1]. According to the Worldometer live statistics, as of 06:50 GMT, 23 June 2022, there have been 546,710,803 cumulatively confirmed coronavirus cases globally, including 6,345,918 deaths and 18,005,205 active cases. More importantly, there is no signs of mitigation or decline in the new cases of the pandemic. The consequences of the COVID-19 pandemic are stretching far beyond the spread of the disease. The tremendous panic, future uncertainty, and a series of urgent COVID-19 prevention measures in early stage such as lockdown, stay-at-home order, mass quarantine, and transport halt, have triggered the worst global economic crisis in a century. The global demands for common manufactured goods and agricultural goods have declined remarkably. The significant consequences of a sharp decline in demand for manufactured goods and services, and extensive containment measures have affected employment and livelihoods of millions of people [2]. In addition, livelihoods of households engaged in informal services such as owners of small grocery stores, day labourers, housekeepers, are also most impacted due to unemployment [3]. While many people with jobs that could be working at home through internet would have fewer negative impacts by the pandemic. Thus, the livelihood impacts of the pandemic are different for different households. Moreover, it is widely reported that the poor, such as landless laborers, wage earners, and small-scale farmers, are most affected due to high livelihood vulnerability [3, 4]. Moreover, the COVID-19 outbreak not only decreased the employment and income of the low-income households, but also increased related costs owing to anti-epidemic products (e.g., face mask, disinfectant, safety goggles), high-price daily products during lockdown period, additional costs of online education, etc. Owing to high spatial heterogeneity in severity of pandemic and livelihood portfolios, empirical studies based on different nations and regions should be conducted.

Livelihood impacts of COVID-19 are dynamic and fluctuant. Owing to the evolving epidemic situations, the livelihood impacts of COVID-19 are dynamic in different countries. In many countries, such as China and South Korea, most regions have returned to normal with schools and offices reopened after two months' lockdown and travel restrictions. Containment measures were evolving based on the severity of virus outbreak and spread. Therefore, the livelihood impacts have a periodical characteristic. Until now, studies evaluating the dynamic influence of the COVID-19 pandemic are limited. Current research mainly focused on the appraisal of livelihood impacts on the early stage of the COVID-19 [4–6]. However, understanding the periodic livelihood impacts of the COVID-19 is critical to developing public health policies during different stages of COVID-19. Thus, comprehensive assessment of the livelihood impacts in different stages should be propelled.

There are mainly three survey methods to generate data that does represent the entire population in COVID-19 pandemic research: telephone interviews, online surveys and face-to-face interviews. Each survey method has its advantages and disadvantages. First, telephone survey has been proven as a successful method which is widely utilized in previous studies [7]. Many studies on Chinese pandemic livelihood impacts based on telephone interviews have been conducted owing to some advantages, including no-contacting interaction, good geographical coverage, ease of use an lower cost compared to face-to-face surveys [8–10]. However, this method also has some disadvantages, such as interviewer bias, lower response rate, missing respondents without access to telephones, the inability to use visual help, unwillingness to display sensitive information [11, 12]. Second, right after the outbreak of COVID-19 in China, a lot of studies based on online surveys have been conducted by virtue of its remarkable advantages, such as nearly no risk of infection, cost-effectiveness, higher speed, higher response rates, large sample size, and flexibility [13–15]. But weak representativeness and low data accuracy are its fatal problems [7, 8]. Third, face-to-face surveys have been justified as an accurate and feasible approach, which are widely used in rural studies by virtue of many advantages, including random sampling, personal interaction, higher accuracy, higher response rates, flexibility, gaining more sensitive information, inclusion of these vulnerable groups(such as the poor, the aged, illiterate) [7, 12]. Face-to-face interviews also have some disadvantages, such as mall sample size, lower cost-effectiveness, geographical limitations, and interviewer bias [16, 17]. However, given the complicated questions and low education of rural interviewees, face-to-face surveys still deliver the most representative results [7]. However, less pandemic livelihood impact research based on in-person surveys have been conducted due to travel restrictions and contagious risk of in-person investigation [18, 19].

The COVID-19 pandemic in China has experienced a whole cycle from epidemic outbreak and strict containment, recovery of working and production, and normal containment. The process of epidemic outbreak and spread in China could be divided into three stages: the outbreak period, recovery period of working and production, and normalized containment period. Governmental countermeasures are changing based on the epidemic situations. The outbreak period in China was from its first report in the end of December 2019 to the mid-March 2020. The interventions in the outbreak period generally include complete lockdown of cities/villages, travel restrictions, rapid investments in increased testing capacity, active case surveillance, treatment of severe cases, isolation of cases, quarantine of cases and high-risk groups, the extension of the Spring Festival holiday, the postponed spring semester for schools, and cordon sanitaire [20, 21]. While in the stage of recovery of working and production from mid-March to the end of April, 2020, the containment measures changed into conditionally re-opened factories (all workers passed the COVID-19 test before working) and businesses (e.g., forbidding dine-in but permitting take-out for restaurants) while travel restrictions were loosened. By early May, 2020, most Chinese cities have returned to normal with schools and offices reopened. The governmental measures changed to increasing COVID-19 tests, encouraging vaccinating, restricting travelling from high-risk regions, behavioral risk-reduction strategies, such as the compulsory use of masks in the public. Therefore, China is an ideal study case to illustrate the dynamic livelihood impacts of the COVID-19 pandemic.

This study investigated the dynamic livelihood impacts of the COVID-19 pandemic in China based on a case study in mountainous areas of China. The main objectives include: (1) the overall livelihood impacts for rural households with different livelihood strategies in the whole year of 2020; (2) the dynamic livelihood impacts of COVID-19 pandemic. The heterogeneity of rural households and dynamics have been highlighted in the study.

## Materials and methods

### Study area

*Xunwu County* is located in the southeast of *Jiangxi Province*, Southeast China (24°30 '40 " to 25°12' 10" N, 115°21 '22 " to 115°54' 25" E). It has a subtropical monsoon climate with high mean precipitation (1650 mm year$^{-1}$) and warm annual temperature (a mean of 18.9°C). *Xunwu County* is a typical mountainous agricultural county in southern China. The terrain is primarily dominated by mountains and hills, while the cropland is insufficient. It is a typically mountainous county where the acreage of mountainous areas accounts for 75.6% of the total. Statistical data indicated that cropland per capita of rural population was 0.058 ha person$^{-1}$ in 2020, and the insufficient cropland substantially hindered the development of crop farming [22]. However, the favourable conditions for growing fruits, such as moderate temperature, sufficient rainfall, abundant upland and large temperature difference between day and night, have promoted fruit growing to become the pillar industry of the county. The primary fruits include navel orange, mandarin orange, and passion fruit. The planting area of navel orange and mandarin orange was 20353.40 ha, which accounted for 87.36% of the total orchards in the end of 2019 [23]. The tremendous planting area and output make *Xunwu County* to gain several titles—"China's Mandarin Orange Hometown", "China's Navel Orange Hometown", and "Export Base of China's Navel Orange". As a result, orange farming become the primary income source of many rural households. In addition, paddy farming and poultry are minor agricultural sectors in rural areas. Owing to relatively lower incomes in primary industry and the underdeveloped secondary industry, the rural-urban migration of young and middle-aged labours is prevalent in the county. More than 320,00 people migrated out the county in the end of 2019, and the net migrating rate was 9.78% in 2019. The primary migrating destinations are developed coastal regions, such as *Guangdong Province*, *Zhejiang Province* and *Fujian Province*. A variety of household livelihood strategies have been developed and livelihood activities are diversified.

### Data collection

The survey data were collected from February 1, 2021 to March 25, 2021. Our surveys were mainly conducted in February, 2021, and follow-up surveys were continued to supplemented some missing and unclear information in March 2021. The primary field survey period coincided with China's Lunar New Year holiday, when a majority of the population returned to their rural homes to celebrate with family. It was just one year after COVID-19 outbreak, and it was easy to recall the recent year's family livelihood information. So it is the most appropriate time to investigate the livelihood impacts on agricultural production, non-farm employment, and daily life based on a whole year. Selection of respondents proceeded according to a two-stage cluster sample. In the first stage, two villages (*Huaxing* and *Shangping*) were randomly chosen. In the second stage, 10% of the rural households in each village were selected randomly for investigation after an inventory of households was enumerated. As a result, 56 households and 47 households were randomly selected for surveying in *Huaxing village* and *Shangping village*, respectively. We also chose two village leaders from each sampling village for in-depth interviews. We conducted about two-hour face to face in-depth interviews with four village leaders. Village-leader questions were designed for open answers. Through these interviews, the heterogeneity and severity of livelihood impacts of COVID-19 for different rural households and different periods could be clarified.

It should be noted that there are many factors influencing our sample size: travel limitations and high risk of infection, high homogeneity and financial limitations. First, strict disease-

control measures (including restrictions of trans-regional travels, reporting travel plan to governments) have been still adopted during our survey period in order to prevent the spread of COVID-19. It is not easy to gain the travel permission if you have not social networks there. And the potential risk of infection was higher because our investigators need to survey many interviewees who may just returned from medium-high risk areas. Second, our pilot survey indicated that there was high homogeneity in livelihood impacts among the same type of rural households. Third, the survey cost per respondent was very high owing to additional costs. However, the small sample size basically could not affect the research results because of qualitative research and high quality of survey data.

Sampled households were interviewed with a questionnaire that included 57 questions. The survey questions of the questionnaire mainly include three parts: basic information of the rural households, COVID-19 impacts on production, and COVID-19 impacts on daily life. The survey was carried out by our professional investigators through a question-and-answer format to ensure the accuracy of the survey. Most questions were answered by householders, the other family members complemented some questions. Follow-up interviews were conducted with select respondents to assure survey reliability and to provide explanations for unclear responses. After eliminating 8 invalid records, we gained 95 effective questionnaires. The effective rate of the questionnaires is 92.23%. The characteristics of these rural households are listed in Table 1. On average, the income per capita of rural residents in these sampling villages was 1798 USD in 2020, slightly below the county rural average of 1896 USD per capita [22]. The fruit plantation per rural registered population was 0.093 ha person$^{-1}$ for our sampling households, which was similar to the county average of 0.089 ha person$^{-1}$. These results implied that two sampling villages could represent the villages of the whole county. Finally, we input all of the data from the questionnaires and analyzed the links between the rural livelihoods and the COVID-19 pandemic.

The ethical standards and principles were followed throughout this study. Ethical approval for this study has been obtained from the Ethical clearance committee of Foshan University, People's Republic of China. Then approaching to the respondents, we explained the purpose of the study and assured them the anonymity, confidentiality and privacy of the responses. All research respondents voluntarily agreed to participate in the research without any pressure. Consent was obtained from all respondents through verbal agreement.

To clarify the livelihood impacts of the COVID-19 for different rural households, the surveyed households were classified into four types according to the proportion of nonfarm income to total: pure farm households (PFH) with nonfarm income proportion less than 10%, mixed farm-business households (type I) (MFH1) with nonfarm income proportion between 10 and 50%, mixed farm-business households (type II) (MFH2) with nonfarm income proportion between 51 and 90%, and nonfarm (NH) households with nonfarm income proportion more than 90%. Then, all the investigated households were stratified for further analysis according to the livelihood non-agriculturization (Table 1).

**Table 1. The characteristics of sampling rural households.**

|  | PFH | MFH1 | MFH2 | NH |
|---|---|---|---|---|
| Sample size (household) | 10 | 37 | 31 | 17 |
| % of total households (%) | 10.5 | 39 | 32.6 | 17.9 |
| Household size (person household$^{-1}$) | 5.2 | 4.6 | 5.4 | 4.4 |
| Net income per farmer (RMB farmer$^{-1}$ year$^{-1}$) | 10131 | 11976 | 14728 | 16410 |

Note: Data in value terms are calculated at current prices; PFH = pure farmer household,

MFH1 = mixed farm-business household (type I), MFH2 = mixed farm-business household (type II), NH = Nonfarm household.

To illuminate the dynamics of the livelihood impacts of the COVID-19 epidemic, the process of livelihood impacts in *Xunwu County*, *Jiangxi Province* in 2020 could be divided into three stages: the outbreak period (from January 23, 2020 to March 11, 2020), recovery period of working and production (from March 12, 2020 to April 30, 2020), and normalized containment period (from May 1, 2020 to the end of 2020). Dynamic governmental countermeasures are mentioned in the above part.

## Results

The livelihood impacts of COVID-19 pandemic could be implemented through three channels: virus encroachment and health problems, governmental containment measures (esp. strict countermeasures in early stage, such as lockdown of cities and villages), and economic decline in the world. Owing to being far away from the city of *Wuhan*, *Hubei Province*, there was no confirmed cases in *Xunwu County* in 2020. Thus, the health impacts of COVID-19 pandemic were not discussed in the study. The primary livelihood influences were derived from the direct disturbance of strict containment measures, and gloomy macro economy in the globe (Fig 1). The former mainly refers to the periodical suspension of most productive and living activities owing to the urgent and extreme countermeasures in early stage, such as lockdown, stay-at-home order, mass quarantine, and transport halt, etc. While the latter means the medium and long-term impacts on global employment and consumption.

### Livelihood impacts on different rural households

**Income loss.** These urgent and extreme countermeasures conducted by governments, esp. lockdowns and trip halt, have significant and negative impacts on rural livelihoods by decreasing households' incomes. Lockdowns are protecting rural people from catching COVID-19, but the economic burden is heavy. Based on established schedule, the holiday of 2020 Spring Festival was from January 24 to January 30 2020, and the starting working day

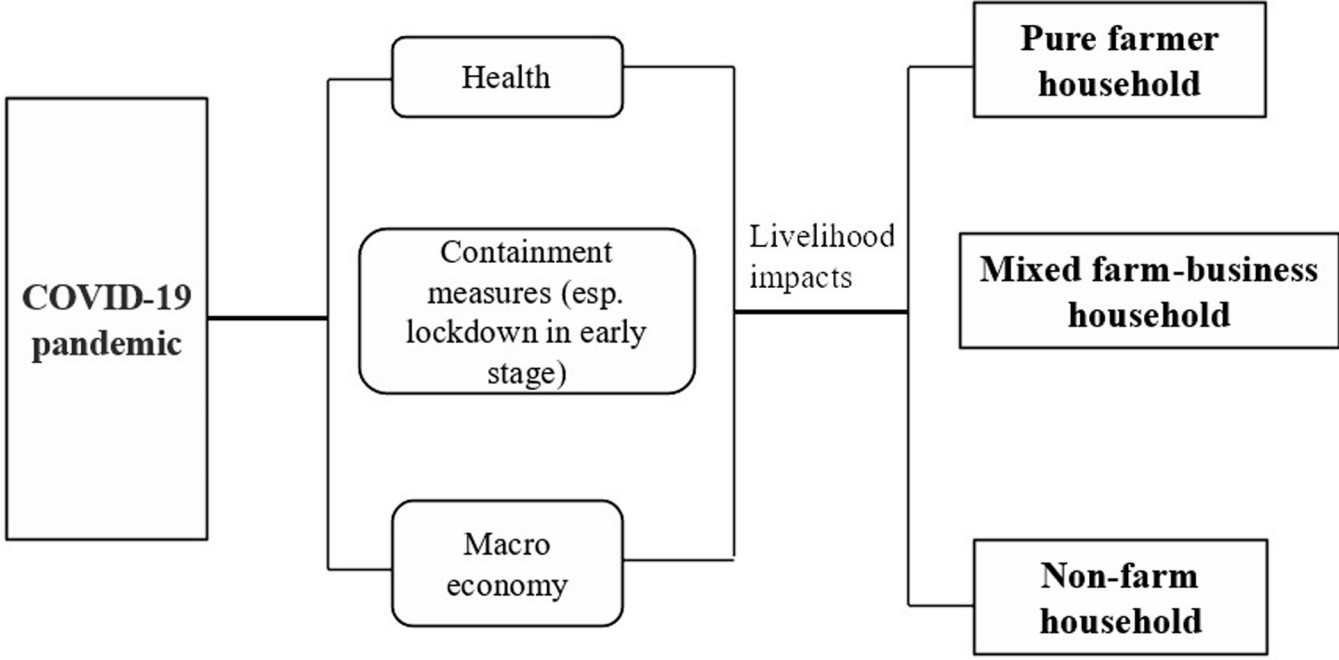

**Fig 1. Mechanism of the livelihood impacts of COVID-19 pandemic on different rural households.**

should be February 1 2020. However, the sudden pandemic has postponed the normal production of most manufacturing factories in *Jiangxi Province* at least 40 days. Since March 12 in 2020, the major public health emergency response in *Jiangxi Province* had adjusted from the provincial first-level to the provincial second-level. The adjustment of countermeasures means that the governmental priority of work changes into recovering the economic operation under the condition of further pushing the prevention and control of the epidemic. Based on our survey, average income loss of all survey households is 6842 RMB, accounting for 13.01% of the total household income in 2020. As for different household types, the overall household income loss for each household of PFH, MFH1, MFH2, and NH is 1800 RMB, 6000 RMB, 9032 RMB, and 9647 RMB respectively in 2020. The economic loss to the total household income of PFH, MFH1, MFH2, and NH is 3.74%,11.67%,12.41% and 17.92% respectively (Table 2). The income loss for PFH is the lowest. Based on our in-depth interviews, the coincidence of the pandemic outbreak period and the slack farming season (from the end of January to mid-March for growing citrus and paddy) determined the weak intensity of income impacts for PFH. The marketing of navel orange in the pandemic outbreak period (late winter and early spring) came to the end and the spring orchard management did not start during the outbreak period. Thus, most sampled PFH (80%) have reported that the epidemic has nearly not affected their agricultural production. But 20% of sampled PFH with a lot of orange stocks have suffered from 10–16% of income loss owing to over-supply and lower farm gate prices of fruits resulting from COVID-19 restrictions on the movement of traders from outside the county and transportation interruption. In a word, along with increasing proportion of non-farm income, the absolute value and relative proportion of household income loss are all increasing.

**Disorders in daily life.** The pandemic impacts on households' daily life mainly occurred in the outbreak period because the strict restrictions on the movement of population and goods. The disorders in daily life induced by the epidemic primarily include: increase of food price and additional expenditures, movement restrictions, restrictions on family parties, and closure of schools. First, during the outbreak period, sudden lockdowns and strict restrictions on movement of goods and population directly led to the scarcity of goods supplies, impeded marketing channels and subsequent rise of food price. Most respondents reported that the retail price of vegetables during the outbreak period generally increased by 20–100%, and the retail price of many imperishable and low-price vegetables (such as potatoes, cabbage, radish) may increase by 2–3 times. The price of meat, such as pork, and beef, increased by about 20% and reached the price peak of history. Though rural households usually grew some vegetables to consume for the left-behind family members, yet the supply could not meet all the family members when migrated members coming back to spend the Spring Festival holiday. The rise of food price not only resulted in additional expenditures, but also resulted in decline of living standards through consuming less high-price foods. In addition, the epidemic also added some additional costs for buying protection products, such as face masks and disinfectant. The specifically additional costs for different rural household could be seen in Table 3.

The outbreak of the epidemic also brought about travel restrictions and restrictions of visiting relatives and friends (esp. the forbidding of mass gathering). Different rural households

**Table 2. Income loss of different rural households.**

|  | PFH | MFH1 | MFH2 | NH | All |
|---|---|---|---|---|---|
| Income loss (RMB) | 1800 | 6000 | 9032 | 9647 | 6842 |
| % of total income in 2020 | 3.74 | 11.67 | 12.41 | 17.92 | 13.01 |

**Table 3. Additional costs of rural households deriving from COVID-19 epidemic (RMB).**

| Household | Protective cost | Additional food cost | Other cost | Total | % of total household income |
|---|---|---|---|---|---|
| PFH | 160 | 210 | 100 | 470 | 1.03 |
| MFH1 | 235 | 420 | 180 | 835 | 1.62 |
| MFH2 | 280 | 860 | 350 | 1490 | 2.05 |
| NH | 360 | 1200 | 500 | 2060 | 2.85 |

have different perceptions about the livelihood impacts on travel restrictions and party restrictions (Fig 2). Generally, NH has relatively high recognition in two aspects compared to other rural households, though the containment measures in two survey villages were the same to all households. However, NFH1 has the highest perception on additional costs (Fig 2).

The intervention of the epidemic on education was profound and long-lasting, esp. for students of elementary schools and middle schools. The spring semester originally planned to start from mid-February of 2020. However, the abrupt epidemic directly led to the closure of all schools, including kindergartens, elementary schools, middle schools and universities. Most school education (except kindergartens) transferred to the internet through cellphones or personal computers. Most parents reported that the study efficiency and effects of online study were extraordinarily poorer than that of school education. The elementary schools, middle and high schools in Jiangxi Province have restored to normal since May 7, 2020 and have conducted just two months' school education. Then schools started the summer holiday from July 6, 2020. Except MFH1, more than half of other cohorts of rural parents complained about the low efficiency and poor effects of online studies. The difference among various types of farmers may indicate the variations of parents' expectations and requirements for children education.

## Different coping strategies in different stages of the pandemic

As for the dynamic livelihood impacts of COVID-19, our survey showed that all sampled households reported that the livelihood impacts of COVID-19 were different along with

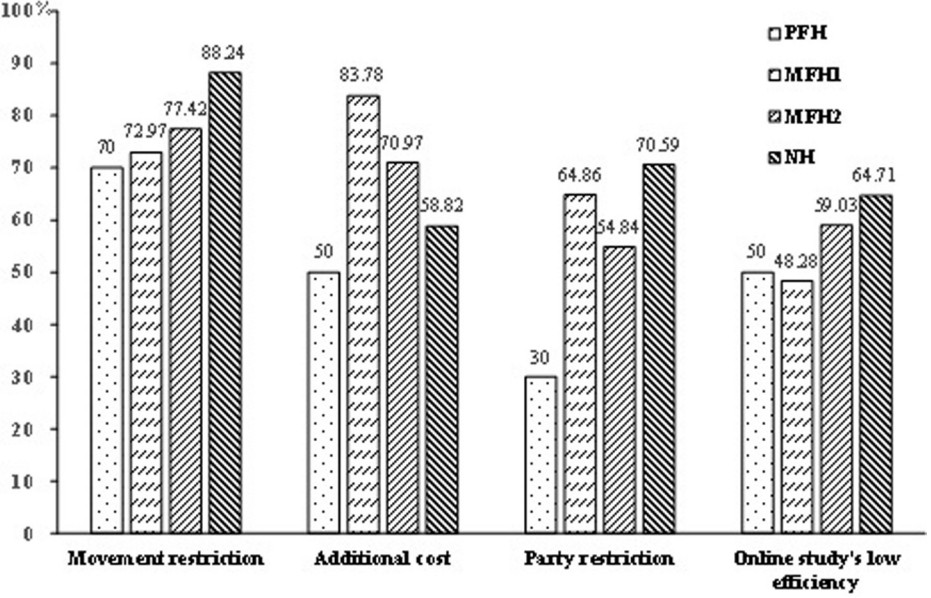

**Fig 2. Perception of COVID-19 impacts on daily life for different rural households.**

evolution of the epidemic situations. To clearly reveal the dynamics of livelihood impacts, the change of coping strategies in different stages is a proper indicator. For different rural households, the livelihood portfolios in different stages were different (Table 4). For PFH, local farming is the primary livelihood. The livelihood fluctuation in different stages is weak because previously tremendous investment in orchard management restricts livelihood transformations. Local odd jobs were important income supplement of PFH, but the strict countermeasures (such as lockdown) in outbreak periods contributed to the low value. Rural-urban migration restarted during the normalized containment period. For MFH1, local farming, local odd jobs and non-local employment were their dominating livelihood strategies. The evolving trends were similar to that of PFH. While internet marketing of agricultural products, decreasing investment and borrowing money are mainly urgent responses to COVID-19 epidemic during outbreak period. For MFH2, the changes of livelihood portfolios were similar to that of MFH1, and the discrepancy mainly lied in the extent of various changes. For NF, going out as migrant workers was the principal livelihood strategy. The recovery process was prominent, and the non-local employment increased from 17.65%, to 41.18% and 64.71% in three stages receptively. Borrowing money and starting up new business in recovery period and normalized containment period were the characteristics of NF. Others mainly include working at home and investment, which are conducted by NF. Specific livelihood strategies for four types of rural households could be seen in Table 4.

## Discussion

The COVID-19 pandemic has brought about huge socioeconomic and livelihood impacts all over the world. The report of the United Nations revealed that the COVID-19 pandemic caused the global economy to shrink by 4.3% in 2020 [24]. The magnitude of economic decline was far higher than the 1.7% reduction during the Great Recession of 2009. The economic shock on China is also tremendous. The epidemic resulted in the economic growth in the first quarter of 2020 is -6.8%, and 2.3% in the whole year of 2020, which was far lower than that of 2019 (6.0%) [25]. The containment measures, such as the lockdowns, quarantine measures and social distancing, contributed to protect people from infecting the disease but disrupted the livelihoods of millions of populations [26]. The economic burden of these strict control measures is remarkably heavy [9, 10]. Research shows that the pandemic's livelihood impacts has tilted towards the most vulnerable population, esp. poor rural households [27]. It is reported that the pandemic could lead to millions of livelihoods are at risk, and an additional 130 million people into extreme poverty [24]. Vulnerable groups including small-scale

**Table 4. Livelihood strategies of different types of rural households in response to the epidemic in various periods (%).**

| Livelihood strategies | PFH | | | MFH1 | | | MFH2 | | | NH | | |
|---|---|---|---|---|---|---|---|---|---|---|---|---|
| | Stage 1 | Stage 2 | Stage 3 | Stage 1 | Stage 2 | Stage 3 | Stage 1 | Stage 2 | Stage 3 | Stage 1 | Stage 2 | Stage 3 |
| Local farming | 90 | 80 | 80 | 75.3 | 70 | 62 | 60.4 | 54.5 | 44 | 11.76 | 5.88 | 0 |
| Local odd job | 30 | 40 | 40 | 40.54 | 50.05 | 56.76 | 32.26 | 41.94 | 45.16 | 29.41 | 41.18 | 41.18 |
| Non-local employment | 0 | 0 | 20 | 18.92 | 29.73 | 40.54 | 22.58 | 32.26 | 48.39 | 17.65 | 41.18 | 64.71 |
| Internet marketing of agricultural products | 10 | 10 | 10 | 10.81 | 2.70 | 2.70 | 10.35 | 8.45 | 4.90 | 0 | 0 | 0 |
| Decreasing investment | 0 | 0 | 0 | 8.11 | 2.70 | 2.70 | 6.45 | 3.23 | 3.23 | 0 | 5.88 | 5.88 |
| Borrowing from friends or banks | 10 | 0 | 0 | 2.70 | 0 | 0 | 3.23 | 0 | 0 | 0 | 11.76 | 11.76 |
| Start up business | 0 | 0 | 0 | 0 | 0 | 0 | 0 | 0 | 0 | 0 | 11.76 | 11.76 |
| Others | 0 | 0 | 0 | 0 | 0 | 0 | 3.23 | 0 | 0 | 17.65 | 5.88 | 5.88 |

*Note*: Stage 1 means the outbreak period, stage 2 means recovery period of working and production, stage 3 means normalized containment period.

farmers, landless farmers, migrant laborers, would be severely affected because of the disruption of their dependent day-to-day work [4]. However, the heterogeneity of rural households and the dynamics of livelihood impacts are generally less discussed in previous studies.

Our results showed that the heterogeneity of rural households was a significant feature of livelihood impacts of COVID-19 epidemic. That is to say, the livelihood impacts of the epidemic are different for different rural households. The household income loss was increasing accompanied by the incremental non-agriculturalization of livelihoods. The income loss of NH is the highest, while it is the lowest for the PFH. The primary cause of the difference in livelihood impacts is whether the epidemic outbreak leading to the interruption of production and employment. The rural households with agriculture-based livelihoods were least adversely affected because the period of the COVID-19 outbreak was coincident with the farming slack seasons. While the unplanned and sudden imposition of the lockdown from March 25 to May 31 2020 in India resulted in a massive and unprecedented disruption to agricultural activities such as harvesting, sale of agricultural produce, and purchase of inputs [28]. The COVID-19 lockdown impact on India agriculture and rural economy was tremendous. While the rural households engaged in employment in secondary sectors and tertiary sectors in our study area were most adversely affected because the disruption in the movement of population and goods during the outbreak period of the epidemic [8, 9]. Some previous studies also revealed the household heterogeneity of livelihood impacts, but they generally highlighted that the heterogeneity of livelihood impacts from gender and land ownership dimensions. That is, the livelihood impacts of female-headed households, female groups and landless farmers were more pronounced, such as food shortage [28–30]. In addition, households with large members or lower educational attainments of householders were also adversely affected [26]. 11 Spatial heterogeneity and industry heterogeneity of livelihood impacts caused by COVID-19 has also been discussed in previous studies [31].

Our results proved that the average household income loss was 6842 RMB and 13.01% of the total in 2020. Moreover, the income loss is different for different rural households, ranging from 1800 RMB to 9647 RMB in absolute value, and the correspondent relative income loss ranging from 3.74% to 17.92%. Especially, the lockdown resulted in a sharp decline in total household income during the outbreak period. The sharp decline of non-farm income is the primary reason. Many small and medium-sized enterprises had to be closed owing to COVID-19 outbreak [8]. This is consistent with the previous observation in China that high rates of unemployment and falling household income are the mainly negative economic outcomes [9, 14, 19, 31, 32]. Our finding is also consistent with past research in many other developing countries, such as India, Ghana, Uganda, and Pakistan [4, 26, 33, 34].

Besides the non-farm sectors, the influence of COVID-19 on agricultural economy is also non-negligible. Farmers' livelihoods are also adversely affected by COVID-19 mainly through disturbing crop production, agricultural products supply, and livestock production and restraining market demands [32, 35, 36]. Especially, the magnitude of pandemic impacts on agricultural production extraordinarily depend on coincidence of lockdowns and key agricultural activities (such as sowing, harvesting, marketing) and external market demands. Previous research indicated that the pandemic impacted almost all stages of the agricultural supply chain but had a greater impact on the sales stage [37]. However, less studies conducted the livelihood appraisal based on a full epidemic cycle, from initial outbreak, recovery, and new normal period.

The pandemic has brought about a series of social disorders, such as increasing food price, movement restrictions, restrictions on family parties, and mental health. More importantly, the additional costs deriving from price rise of food and consumption of protective products not only led to increasing expenditures, but also may lead to transient decline in living

standards. These results were consistent with previous studies of China that COVID-19 caused a lot of negative social outcomes for rural households, such as rising prices, healthcare inaccessibility, aggravating poverty, and undermining mental health [9, 10, 14, 15]. Many studies in poor countries, such as Uganda, India, showed that the Covid-19 lockdown directly caused decreasing food intake, decreasing nutrient-dense foods, simplification of food, and even starvation [30, 34, 38]. The extent of poverty and duration of lockdown are the root causes of the research differences.

The pandemic has caused the closure of universities and elementary schools in many countries. The impacts on children education for rural households are more far-reaching. Though the education could transform into online teaching, learning by the internet, radio and television, yet the effect of online learning for rural children was poor owing to lacking better learning devices (such as computers), internet connectivity, less monitoring, poor learning environment as compared to urban students [9, 39]. Previous studies also found that closures of schools and day-care facilities have resulted in an increase in the burden of women, uneven access to home schooling for girls, increasing the poverty levels of females [24]. And there was just about a half attendance of home schooling, despite the free supply of government-sponsored educational television, radio and digital programmes [40].

Our study also revealed that there were different livelihood responses in different stages of the epidemic. Along with the evolution of the epidemic, responses in livelihood strategies to the threat of COVID-19 were different in various stages. In the outbreak period, many rural households' livelihood strategies transformed into local farming, marketing of agricultural products through internet, and conducting local casual jobs, to adapt to the lockdowns. Our results were similar with previous studies, which acknowledged the dynamic livelihood impacts the COVID-19 and focused on the impact assessment of the COVID-19 lockdown [9, 10, 28, 41].

## Strengths and limitations

Our study has some major strengths. Firstly, perhaps this is the first survey that evaluates the systematic impact of COVID-19 pandemic lockdown on rural households' livelihoods in China based on a whole year as the study period. The dynamic livelihood impacts have been discussed by virtue of inclusion of complete pandemic cycle from initial outbreak, to intermediate recovery and finally new normal stage in 2020. Secondly, traditional face-to-face interviews have been utilized to clarify the specific livelihood impacts of rural households. Timely survey just one year after COVID-19 outbreak was achieved after we overcame many difficulties. Finally, the household heterogeneity of the livelihood impacts have also been highlighted in our study. This study also has several limitations. First, sample size, including the sampled villages and respondents, was not enough to represent all villages of China. Second, spatial heterogeneity of livelihood impacts has not been discussed owing to data limitations. Finally, all the field survey data was self-reported and was based on subjective estimation.

## Conclusions and policy implications

The COVID-19 had a strong impact on developing economies, leaving the biggest negative effects on rural households, which are the primary part of the poor in the world. The outbreak of the pandemic and containing measures and restrictions adopted by various countries are likely to intensify the poverty intensity by decreasing employment opportunities and household incomes, increasing the additional expenditure deriving from remarkable rise of food price and protection cost, reducing living standards, etc. The average income loss of all survey households is 6842 RMB, accounting for 13.01% of the total household income in 2020. And

the income loss was different for different rural households. The household income loss was increasing accompanied by the increasing non-agriculturalization of livelihoods because lockdowns caused more than two months' interruption in most secondary sectors and tertiary sectors of China. While the coincidence of the pandemic outbreak period and the slack farming season is the primary cause of the lower loss of agriculture. The pandemic has also brought about a series of disorders in daily life, such as rise of food price and additional expenditure, travel restrictions, restrictions on family parties, closure of schools and deceasing living standards. The impacts of daily life among various rural households are also different. In general, the NFs' perceptions of the disturbance on daily life are the highest compared to other types of rural households. Our results showed that the livelihood impacts of the pandemic were dynamic. Along with the evolution of the epidemic, responses in livelihood strategies to the threat of COVID-19 were different in various stages.

Our study has some policy implications. First, severe lockdown and quarantine measures should be more precisely and cautiously used because these measures have produced significant livelihood loss for most rural households. Second, supportive policies for rural households should considered the heterogeneity of rural households. Third, promoting farmers' livelihood resilience should be placed as the priority of local governments. Many governmental policies, including diversified farming, skill training, internet marketing, and share of employment information should be encouraged in rural areas. Finally, farmer's cooperative organization should be promoted to cope with external uncertainty.

## Supporting information

**S1 Data.**
(XLSX)

## Author Contributions

**Funding acquisition:** Chengchao Wang.

**Investigation:** Xiu He.

**Writing – original draft:** Chengchao Wang.

**Writing – review & editing:** Chengchao Wang, Xianqiang Song, Shanshan Chen, Dongshen Luo.

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
