## [Decision Letter · Decision Letter 0]

6 Jun 2022

PONE-D-22-11264Dynamic Livelihood Impacts of COVID-19 on Different Rural Households in Mountainous Areas of ChinaPLOS ONE

Dear Dr. Wang,

Thank you for submitting your manuscript to PLOS ONE. After careful consideration, we feel that it has merit but does not fully meet PLOS ONE’s publication criteria as it currently stands. Therefore, we invite you to submit a revised version of the manuscript that addresses the points raised during the review process.

 The paper is certainly addressing a topical issue related to the global pandemic in rural households in China. I agree with Reviewer 1 that it carries novelty, thus, has the potential to add to existing literature. However, there are numerous fundamental shortcomings the author must devote more time to. Reviewer 2 is right in pointing out that the paper overgeneralizes the research results (2 villages looked at vs. 700,000 villages in the country). It is therefore highly recommended to introduce and detail this work as a case study, adding the right methodology explanation, too. The scientifically sound and correct contextualization of the sample is crucial. Comments 4 and 5 of Reviewer 2 are also important to properly respond to. You need to expand your lit review and include many more relevant papers on rural China - this was also underscored by Reviewer 3. All other comments made by Reviewer 3 must be dealt with seriously in a major revision. In sum, I do not support to reject the paper – although it is close to it – but rather ask you to add more precious time to thoroughly review and improve the manuscript.

We look forward to receiving your revised manuscript.

Kind regards,

István Tarrósy, PhD

Academic Editor

PLOS ONE

Journal Requirements:

5. PLOS requires an ORCID iD for the corresponding author in Editorial Manager on papers submitted after December 6th, 2016. Please ensure that you have an ORCID iD and that it is validated in Editorial Manager. To do this, go to ‘Update my Information’ (in the upper left-hand corner of the main menu), and click on the Fetch/Validate link next to the ORCID field. This will take you to the ORCID site and allow you to create a new iD or authenticate a pre-existing iD in Editorial Manager. Please see the following video for instructions on linking an ORCID iD to your Editorial Manager account: https://www.youtube.com/watch?v=_xcclfuvtxQ.

6. We note that Figure 1 in your submission contain [map/satellite] images which may be copyrighted. All PLOS content is published under the Creative Commons Attribution License (CC BY 4.0), which means that the manuscript, images, and Supporting Information files will be freely available online, and any third party is permitted to access, download, copy, distribute, and use these materials in any way, even commercially, with proper attribution. For these reasons, we cannot publish previously copyrighted maps or satellite images created using proprietary data, such as Google software (Google Maps, Street View, and Earth). For more information, see our copyright guidelines: http://journals.plos.org/plosone/s/licenses-and-copyright.

a) You may seek permission from the original copyright holder of Figure 1 to publish the content specifically under the CC BY 4.0 license.  

Reviewers' comments:

Reviewer's Responses to Questions

**Comments to the Author**

1. Is the manuscript technically sound, and do the data support the conclusions?

Reviewer #1: Yes

Reviewer #2: No

Reviewer #3: Yes

2. Has the statistical analysis been performed appropriately and rigorously? 

Reviewer #1: Yes

Reviewer #2: Yes

Reviewer #3: Yes

3. Have the authors made all data underlying the findings in their manuscript fully available?

Reviewer #1: Yes

Reviewer #2: Yes

Reviewer #3: Yes

4. Is the manuscript presented in an intelligible fashion and written in standard English?

Reviewer #1: Yes

Reviewer #2: Yes

Reviewer #3: Yes

5. Review Comments to the Author

Reviewer #1: The manuscript is technically solid, the data obtained by the authors supports the investigation and its conclusions, and thus the study exhibits novel research findings: i find it especially important to focus on those households, those individuals, who were not in the center of attention of different government policies, especially the vulnerable groups we should talk about, in order to understand the real consequences of the global pandemic and the lockdowns.

According to the given data, the authors have not previously published this study. The paper's format is logical, i would say a bit short, but the main elements were included. The studies are carried out to a high technical degree and are sufficiently detailed within the publication, demonstrating that the analysis was carried out appropriately and systematically. The authors have included all data supporting their findings in their publication in tables, along with explanations for those tables. Additionally, the conclusion is given in the appropriate manner, the evidence supports the conclusions. The research complies with all applicable ethical and research integrity standards, and the work adheres to appropriate reporting criteria and community standards for data access.

Reviewer #2: In this paper the authors seek to document the impacts of COVID on the rural economy. They claim their objective is to show that there are heterogeneous effects to the pandemic and the measures taken by the government to fulfill it. They look at the impact one year (approximately 12 Months) after the lockdown (which lasted for 2 months strictly / 4 or so months less strictly). To achieve their goal, they used a survey of 95 families in 2 villages in one county in southern China. The survey had 21 questions. The study found negative impacts … and said that on average families suffered 13% fall in income … as well as other impacts (such as higher prices).

Although this is an important topic and the paper is fairly well written, there are a number of serious issues. Here they are below:

Major Comment 1:

The sample size is a severe problem. There are three sub problems:

1a: The study is supposed to look at impacts on rural China. Unfortunately, rural China has 700,000 villages. This study went to 2 of them. This needs to be addressed. It might be a fatal flaw (we really learn nothing about anything more than 2 villages).

1b: The authors do not put their village in the context of other villages. This at least needs to be done.

1c: The authors only survey 95 households and then divide them into 4 groups. This means on average there are less than 25 sample observations per group. I again dear this undermines the convincing nature of the study.

Major Comment 2:

The authors only asked 21 questions. In a standard survey of farm households it takes an entire section of questions to get a good estimate of income. Not to mention the other issues (eg, impact on child’s education; impact on consumer prices; etc). Plus, the authors were comparing to the year before. I have high doubts about the ability of such a data set to cast clear light on the issues at hand.

Major Comment 3:

In the discussion section, the authors compare their results to other countries. This is good. However, we need to know more about the papers and comparisons. More discussion is needed.

Major Comment 4:

There are many papers on rural China that are not cited. For example, there are three papers by Dr. Huan Wang and her team from Stanford that use data from 700 villages and 7 provinces that are published in SCI and SSCI journals that are not cited. This illustrates a problem in their literature review. Have they missed other papers?

Major Comment 5:

There is no set of limitations and strengths in the paper. These are needed.

Reviewer #3: The outbreak of global COVID-19 pandemic has brought about severe negative impacts on livelihoods of the poor. Under this content, this is an interesting work. The study focuses on the impact of the global outbreak of COVID-19 on the livelihoods of vulnerable populations. The MS could provide a comprehensive livelihood appraisal on China. The study is worth being published, but there are some shortcomings.

1. Data on victims of the pandemic need to be updated at line 26-29, Page 1.

2. Introduction: add some recent studies, especially related studies of China.

3. Reasons to select the study case need to be highlighted.

4. Discussion: the spatial difference of the livelihood impacts should be emphasized. Especially, the Shanghai City has experienced 2 months lockdown. If the livelihood impacts and responses of rural households around the city were similar to these of the research.

5. Policy implication of the study should be supplemented in the last section.

6. PLOS authors have the option to publish the peer review history of their article (what does this mean?). If published, this will include your full peer review and any attached files.

Reviewer #1: No

Reviewer #2: No

Reviewer #3: No

---

## [Author Response · Author response to Decision Letter 0]

24 Jul 2022

Reviewer #1:

Comment 1:

The manuscript is technically solid, the data obtained by the authors supports the investigation and its conclusions, and thus the study exhibits novel research findings: i find it especially important to focus on those households, those individuals, who were not in the center of attention of different government policies, especially the vulnerable groups we should talk about, in order to understand the real consequences of the global pandemic and the lockdowns.

 According to the given data, the authors have not previously published this study. The paper's format is logical, i would say a bit short, but the main elements were included. The studies are carried out to a high technical degree and are sufficiently detailed within the publication, demonstrating that the analysis was carried out appropriately and systematically. The authors have included all data supporting their findings in their publication in tables, along with explanations for those tables. Additionally, the conclusion is given in the appropriate manner, the evidence supports the conclusions. The research complies with all applicable ethical and research integrity standards, and the work adheres to appropriate reporting criteria and community standards for data access.

Response:

We are grateful for your constructive suggestion and positive appraisal on our manuscript, we will further improve it to according to the comments. 

Reviewer #2:

Comment 1:

The sample size is a severe problem. There are three sub problems:

1a: The study is supposed to look at impacts on rural China. Unfortunately, rural China has 700,000 villages. This study went to 2 of them. This needs to be addressed. It might be a fatal flaw (we really learn nothing about anything more than 2 villages).

Response:

Thank you for your valuable remind and constructive advice. The representativeness of the samples is the fundamental to survey research. Appropriate sampling methods and a certain scale of sample size is necessary for a national study. 

We randomly selected these two villages, which does not mean that they represent 700,000 villages in China, but only represent common villages which consider fruit farming as the leading industry in mountainous agricultural county in southern China. Through in-depth investigation, some profound viewpoints could be discovered. Some missing information may produce some misunderstanding. We supplemented some important information in “2.1 Study Area” section according to the comment. Please see our revised MS.

Comment 1:

1b: The authors do not put their village in the context of other villages. This at least needs to be done.

Response:

We revised our MS according to the comment.

Comment 1:

1c: The authors only survey 95 households and then divide them into 4 groups. This means on average there are less than 25 sample observations per group. I again dear this undermines the convincing nature of the study.

Response:

Thanks for your important question. Some important explanation was neglected in our MS. We supplemented some information in our revised MS as follows: “It should be noted that there are many factors influencing our sample size: travel limitations and high risk of infection, high homogeneity and financial limitations. First, strict disease-control measures (including restrictions of trans-regional travels, reporting travel plan to governments) have been still adopted during our survey period in order to prevent the spread of COVID-19. It is not easy to gain the travel permission if you have not social networks there. And the potential risk of infection was higher because our investigators need to survey many interviewees who may just returned from medium-high risk areas. Second, our pilot survey indicated that there was high homogeneity in livelihood impacts among the same type of rural households. Third, the survey cost per respondent was very high owing to additional costs. However, the small sample size basically could not affect the research results because of qualitative research and high quality of survey data.” 

In addition, our study was a case study which indicated the livelihood impacts of a mountainous agricultural county with developed fruit farming in southern China. Special agriculture in southern China is very developed in some mountains. The livelihood impacts of COVID-19 were different from these of rural areas mainly planting grain crops. The method of qualitative research has been adopted, which need less surveyed samples.

Major Comment 2:

The authors only asked 21 questions. In a standard survey of farm households it takes an entire section of questions to get a good estimate of income. Not to mention the other issues (eg, impact on child’s education; impact on consumer prices; etc). Plus, the authors were comparing to the year before. I have high doubts about the ability of such a data set to cast clear light on the issues at hand.

Response:

We are appreciated that you identified our errors. “21” should be replaced by “57”. Please see our revised MS. 

Major Comment 3:

In the discussion section, the authors compare their results to other countries. This is good. However, we need to know more about the papers and comparisons. More discussion is needed.

Response: 

Thanks for the comment. We added more discussion to in our revised MS.

Major Comment 4:

There are many papers on rural China that are not cited. For example, there are three papers by Dr. Huan Wang and her team from Stanford that use data from 700 villages and 7 provinces that are published in SCI and SSCI journals that are not cited. This illustrates a problem in their literature review. Have they missed other papers?

Response: 

We revised our MS according to the comment.

Major Comment 5:

There is no set of limitations and strengths in the paper. These are needed.

Response:

We revised our MS according to the comment.

Reviewer #3: 

Major Comment 1:

Data on victims of the pandemic need to be updated at line 26-29, Page 1.

Response:

Thanks for the comment. We have updated data on pandemic victims.

Major Comment 2:

 Introduction: add some recent studies, especially related studies of China.

Response:

We appreciate your constructive comments. We supplemented included some recent literature in the manuscript according to the comment.

Major Comment 3:

Reasons to select the study case need to be highlighted.

Response:

We revised our MS according to the comment.

Major Comment 4:

Discussion: the spatial difference of the livelihood impacts should be emphasized. Especially, the Shanghai City has experienced 2 months lockdown. If the livelihood impacts and responses of rural households around the city were similar to these of the research.

Response:

We revised our MS according to the comment.

Major Comment 5:

Policy implication of the study should be supplemented in the last section.

Response:

We revised our MS according to the comment.

---

## [Decision Letter · Decision Letter 1]

16 Aug 2022

Dynamic Livelihood Impacts of COVID-19 on Different Rural Households in Mountainous Areas of China

PONE-D-22-11264R1

Dear Dr. Wang,

We’re pleased to inform you that your manuscript has been judged scientifically suitable for publication and will be formally accepted for publication once it meets all outstanding technical requirements.

Kind regards,

István Tarrósy, PhD

Academic Editor

PLOS ONE

Additional Editor Comments (optional):

I can see that the author has dealt with the issues and critical points raised by the reviewers and made the manuscript fit for publication.

Reviewers' comments:

Reviewer's Responses to Questions

**Comments to the Author**

1. If the authors have adequately addressed your comments raised in a previous round of review and you feel that this manuscript is now acceptable for publication, you may indicate that here to bypass the “Comments to the Author” section, enter your conflict of interest statement in the “Confidential to Editor” section, and submit your "Accept" recommendation.

Reviewer #1: All comments have been addressed

Reviewer #3: All comments have been addressed

2. Is the manuscript technically sound, and do the data support the conclusions?

Reviewer #1: Yes

Reviewer #3: Yes

3. Has the statistical analysis been performed appropriately and rigorously? 

Reviewer #1: Yes

Reviewer #3: Yes

4. Have the authors made all data underlying the findings in their manuscript fully available?

Reviewer #1: Yes

Reviewer #3: Yes

5. Is the manuscript presented in an intelligible fashion and written in standard English?

Reviewer #1: Yes

Reviewer #3: Yes

6. Review Comments to the Author

Reviewer #1: After reading the submission, the adjustments, and the requests made by the other reviewer, I am certain that the authors have responded to all of the changes. I have no further requests, I believe that the changes have made this submission acceptable - from my side. While I believe the other reviewer's point regarding the number of settlements in China is significant, it should also be noted that data from hundreds of remote villages with varying social, geographical, and climatic characteristics would not contribute to study or clarity.

Reviewer #3: The authors have addressed well with all my suggestiones. I have no other comments. It is suggested to be accepted.

7. PLOS authors have the option to publish the peer review history of their article (what does this mean?). If published, this will include your full peer review and any attached files.

Reviewer #1: No

Reviewer #3: No

---

## [Editor Report · Acceptance letter]

2 Sep 2022

PONE-D-22-11264R1 

Dynamic Livelihood Impacts of COVID-19 on Different Rural Households in Mountainous Areas of China 

Dear Dr. Wang:

I'm pleased to inform you that your manuscript has been deemed suitable for publication in PLOS ONE. Congratulations! Your manuscript is now with our production department. 

Kind regards, 

on behalf of

Dr. István Tarrósy 

Academic Editor

PLOS ONE